# Replicating prediction algorithms for hospitalization and corticosteroid use in patients with inflammatory bowel disease

**Ryan W. Gan[1], Diana Sun[1], Amanda R. Tatro[2], Shirley Cohen-Mekelburg[3,4], Wyndy L. Wiitala[4], Ji Zhu[5], Akbar K. Waljee**[3,4]*

**1** Genentech, Inc., South San Francisco, California, United States of America, **2** Roche Pharma AG, Basel, Switzerland, **3** University of Michigan Health System, Ann Arbor, Michigan, United States of America, **4** Veterans Affairs Health Care System, Center for Clinical Management Research, Ann Arbor, Michigan, United States of America, **5** Department of Statistics, University of Michigan, Ann Arbor, Michigan, United States of America

* awaljee@umich.edu

## Abstract

### Introduction

Previous work had shown that machine learning models can predict inflammatory bowel disease (IBD)-related hospitalizations and outpatient corticosteroid use based on patient demographic and laboratory data in a cohort of United States Veterans. This study aimed to replicate this modeling framework in a nationally representative cohort.

### Methods

A retrospective cohort design using Optum Electronic Health Records (EHR) were used to identify IBD patients, with at least 12 months of follow-up between 2007 and 2018. IBD flare was defined as an inpatient/emergency visit with a diagnosis of IBD or an outpatient corticosteroid prescription for IBD. Predictors included demographic and laboratory data. Logistic regression and random forest (RF) models were used to predict IBD flare within 6 months of each visit. A 70% training and 30% validation approach was used.

### Results

A total of 95,878 patients across 780,559 visits were identified. Of these, 22,245 (23.2%) patients had at least one IBD flare. Patients were predominantly White (87.7%) and female (57.1%), with a mean age of 48.0 years. The logistic regression model had an area under the receiver operating curve (AuROC) of 0.66 (95% CI: 0.65−0.66), sensitivity of 0.69 (95% CI: 0.68−0.70), and specificity of 0.74 (95% CI: 0.73−0.74) in the validation cohort. The RF model had an AuROC of 0.80 (95% CI: 0.80−0.81), sensitivity of 0.74 (95% CI: 0.73−0.74), and specificity of 0.72 (95% CI: 0.72−0.72) in the validation cohort. Important predictors of IBD flare in the RF model were the number of previous flares, age, potassium, and white blood cell count.

**Data Availability Statement:** The Optum EHR [5] data used in this study were licensed from Optum

and are not publicly available due to data licensing and use agreements; interested researchers can contact Optum to license the data. All interested researchers can access the data in the same manner as the authors. The authors had no special access privileges. Optum EHR contact website: https://www.optum.com/business/solutions/government/federal/data-analytics-federal/clinical-data.html. R code (GitHub: https://github.com/CCMRcodes/IBD_Flare) and paper previously published by Waljee et al.[3] was reviewed and adapted to Python code. Manuscript code used to produce tables and plots can be found in the public GitHub repository: https://github.com/phcanalytics/ibd_flare_model. Summary data used to create plots and tables can be found in the public GitHub repository: https://github.com/phcanalytics/ibd_flare_model/tree/master/results. Note, SHAP values require raw data to be calculated and are not included in the repository due to data use agreements. Transparent Reporting of a Multivariable Prediction Model for Individual Prognosis or Diagnosis (TRIPOD) guidelines were implemented;[24] checklist can be found in S3 Table.

**Funding:** Funding to license Optum data was provided by Roche/Genentech. The funder (Roche/Genentech) provided support in the form of salaries for authors Ryan Gan, Diana Sun, Amanda Tatro, and licensing fees for Optum EHR, but did not have any additional role in the study design, data collection and analysis, decision to publish, or preparation of the manuscript. The specific roles of these authors are articulated in the 'author contributions' section.

**Competing interests:** Ryan Gan and Diana Sun are full time employees of Genentech, Inc., a member of the Roche group, and own shares of Roche stock. Amanda Tatro is a full time employee of F. Hoffmann La Roche AG and own shares of Roche stock. This does not alter out adherence to PLOS ONE policies on sharing data and materials. No other authors have competing interests.

## Conclusion

The machine learning modeling framework was replicated and results showed a similar predictive accuracy in a nationally representative cohort of IBD patients. This modeling framework could be embedded in routine practice as a tool to distinguish high-risk patients for disease activity.

## Introduction

Inflammatory bowel disease (IBD), encompassing ulcerative colitis (UC) and Crohn's disease (CD), is a chronic, relapsing, and remitting gastrointestinal disease affecting 1.5 million people in the United States [1, 2]. Although patients may experience periods of remission, both UC and CD can flare up unpredictably, resulting in substantial morbidity, loss of productivity and high medical costs.

Waljee et al. [3] developed a novel machine learning model for predicting hospitalization and corticosteroid use as a surrogate for IBD flares using data from the U.S. Veterans Health Administration (VHA). The primary results from this study found the random forest (RF) model outperformed the logistic regression model's ability to predict IBD flares ([RF]: 79.7% sensitivity and 80% specificity; logistic regression: 64% sensitivity and 64% specificity) [3].

However, the study was performed using VHA data in a predominantly male population, limiting its generalizability. In order to implement machine learning models into practice, replication of model-based predictions in more representative cohorts are necessary. Furthermore, recent advances in machine learning can offer transparency between model output and variables [4], allowing for informed judgment. Ensuring reproducibility and interpretability of machine learning models can increase understanding and usage of these algorithms for clinical decision support.

The objective of this study was two-fold: (1) to replicate the models developed by Waljee et al. [3] to predict IBD flares in a community-based cohort; and (2) to understand which demographic and laboratory data were most informative at predicting IBD flares.

## Methods

### Data source

This was a retrospective observational analysis using data from Optum Electronic Health Records (EHR) Database, which contains de-identified clinical and administrative data from more than 140,000 providers at 700 hospitals and 7,000 clinics [5]. Data are obtained from physician offices, emergency rooms, laboratories, and hospitals and include demographic information, vital signs and other observable measurements, medications prescribed and administered, laboratory test results, administrative data for clinical and inpatient stays, and coded diagnoses and procedures. Data are de-identified in compliance with the Health Insurance Portability and Accountability Act Expert Method and managed according to Optum customer data use agreements.

### Study population

Patients were selected for inclusion if they had at least 2 diagnosis codes (International Classification of Diseases, 9th Revision, Clinical Modification [ICD-9-CM] or ICD, 10th Revision, Clinical Modification [ICD-10-CM]) for UC or CD during at least 2 clinical encounters

between 1 January 2007 and 31 December 2017, with at least 1 clinical encounter being an out-patient visit [6]. For each patient, the date of the first recorded diagnosis of IBD was defined as the index date. Patients were required to have at least 12 months of follow-up after the index date to the last recorded IBD visit in the EHR, and were observed from the index date until the last patient visit, death, or until the end of study (31 December 2018), whichever occurred first. Patients were classified as having UC if all diagnosis codes were 556.xx for ICD-9-CM or K51. xx for ICD-10-CM, CD if all diagnosis codes were 555.xx for ICD-9-CM or K50.xx for ICD-10-CM, and with indeterminate colitis (IC) otherwise. Only patients aged 18 years and older were included in the analysis.

## Outcome measures

The outcome of interest was IBD flare within 6 months of a visit, defined as a composite measure capturing either an IBD-related hospitalization or corticosteroid use for IBD. Inflammatory bowel disease-related hospitalization was considered an inpatient or emergency room admission associated with any diagnosis of UC or CD. Inflammatory bowel disease-indicated corticosteroid use was defined as a prescription for an outpatient oral corticosteroid (S1 List). The indication for corticosteroid prescriptions was determined by searching for diagnosis codes for common inflammatory comorbid conditions (S2 List) within 7 days prior to the corticosteroid prescription date, and excluded those corticosteroid prescriptions associated with a non-IBD diagnosis. Corticosteroid prescriptions fills with a day supply of < 7 days were excluded, following the same approach as Waljee et al. to exclude steroid use for non-IBD indications [3]. Hospitalization or corticosteroid use was assumed to be part of a previous course if it occurred within 90 days of the previous hospitalization or corticosteroid prescription, similar to the approach presented by Waljee et al. [3]

## Predictor variables

Predictor variables included age, sex, race, use of immunosuppressive medication, laboratory data, laboratory summaries (S1 Table), and number of previous IBD flares. Laboratory data included previous laboratory values and laboratory values from the current visit; values that were out of the physiologically possible range were excluded from the analysis. Fecal calprotectin was not included as a predictor variable because it was available in < 1% of the study population. Missing laboratory values were imputed using the population median value. Laboratory data missing in > 70% of the visits were excluded from the analysis.

## Statistical analysis

**Model development.** Three prediction models that looked at the summed contribution of all predictor variables for a given observation were developed to predict IBD flares: (1) a logistic regression using demographic data only, (2) a logistic regression using demographic and laboratory data, and (3) a RF using demographic and laboratory data. The models were assessed at the visit-level, with each patient contributing data from any inpatient or outpatient visits. At each visit, a prediction was made to evaluate whether an IBD flare (defined as an IBD-related hospitalization or corticosteroid use) occurred within the next 6 months.

**Logistic regression using demographic data (LR-Demo).** A logistic regression model using demographic data (S1 Table) was developed. This model aimed to evaluate the predictive ability of demographic data for the risk of hospitalization and corticosteroid use during the 6-month window following every visit.

**Logistic regression using demographic and laboratory data (LR-DemoLab).** A logistic regression model using demographic and laboratory data (S1 Table) was developed to evaluate

the added utility of laboratory data. To avoid overfitting the model, an L1 regularization penalty was added [7, 8]. An optimal λ hyper parameter of 1.11 for the L1 penalty was identified using a 5-fold cross validation. Regularized coefficients were plotted as odds ratios to understand their relationship with odds of a flare in the 6 months following the visit. Laboratory values were standardized after imputation to allow for interpretation of a standard deviation increase.

**RF using demographic and laboratory data (RF-DemoLab).** An RF model using demographic and laboratory data (S1 Table). RF is a tree-based technique that turns a large collection of de-correlated trees into a prediction function [9]. Using demographic and laboratory data, an RF model with 500 trees was fit. Variable importance by Gini impurity were plotted were identify the top 10 variables in the trained RF model. Laboratory values were not standardized in this model to allow for ease of interpretation.

Shapley Additive exPlantation (SHAP) values were estimated using the TreeSHAP algorithm [4] on the trained RF model. The SHAP values quantify the contribution of each variable on the prediction made by the model [4]. The SHAP values are estimated using the withheld validating set, and are the average conditional expected value from a trained tree-based model [4]. Global contributions of the top 10 variables were estimated for the model overall and plotted using a SHAP summary plot. Explanatory plots were created for the top 4 predictor variables.

**Training and validating sets.** To validate the predictive ability of the models, visits were randomly split into training and validating sets, with 70% of the visits directed into the training set and 30% reserved for the validating set. Because flare is a rare event, the non-events were randomly under-sampled to match the distribution of events to avoid class imbalance; model predicted probabilities were corrected to account for the rare event proportion using Bayes' rule [10].

**Model performance.** Model performance was evaluated using the following metrics: sensitivity, specificity, positive predictive value, negative predictive value, accuracy, area under the receiver operating curve (AuROC), and decision curve analysis (DCA) [11]. Brier scores were reported as an overall measure of model performance, where lower values indicate better calibration and discrimination [12]. Non-parametric bootstrap of 100 iterations were performed to estimate model performance 95% CIs.

**Sensitivity analyses.** Sensitivity analyses to evaluate the robustness of the RF-DemoLab model included: (1) sub-populations of patients with UC, CD and IC, and (2) the RF-Demo-Lab model with imputation of laboratory values using multiple imputation by chain equation (MICE) [13, 14].

**Software.** All data processing, cleaning, and analyses were performed in Python 3. Data processing was conducted using Teradata SQL [15], SQLAlchemy [16], Jinja2 [17], and Pandas [18]. Model fitting and preprocessing were conducted using Scikit-learn [19] and Numpy [20]. Random under sampling was performed using Imbalanced-learn [21]. TreeSHAP values calculated and plots created using the TreeSHAP package [4]. Plots were created using Matplotlib [22] and Seaborn [23].

## Patient involvement

No patients were involved in this study as we used de-identified Optum EHR data.

## Data and code availability

The Optum EHR [5] data used in this study were licensed from Optum and are not publicly available due to data licensing and use agreements; interested researchers can contact Optum to license the data. All interested researchers can access the data in the same manner as the

authors. The authors had no special access privileges. Optum EHR contact website: https://www.optum.com/business/solutions/government/federal/data-analytics-federal/clinical-data.html.

R code (GitHub: https://github.com/CCMRcodes/IBD_Flare) and paper previously published by Waljee et al. [3] was reviewed and adapted to Python code.

Manuscript code used to produce tables and plots can be found in the public GitHub repository: https://github.com/phcanalytics/ibd_flare_model. Summary data used to create plots and tables can be found in the public GitHub repository: https://github.com/phcanalytics/ibd_flare_model/tree/master/results. Note, SHAP values require raw data to be calculated and are not included in the repository due to data use agreements.

Transparent Reporting of a Multivariable Prediction Model for Individual Prognosis or Diagnosis (TRIPOD) guidelines were implemented [24]; checklist can be found in S3 Table.

## Results

### Patient characteristics

In total, 454,769 patients with an IBD diagnosis were identified between 2007 and 2017. Patients were excluded if they did not have at least 2 diagnosis codes for UC or CD during at least 2 encounters, with at least 1 encounter being an outpatient visit; did not have at least 365 days between the index date and last recorded visit; less than 18 years old; and did not have any laboratory data. The final cohort consisted of 95,878 patients and 780,559 visits (Fig 1).

Patients were predominantly White (87.7%) and female (57.1%), with a mean (SD) age of 48.2 years (16.8 years) at the time of index (Table 1). The median (IQR) follow-up time for all patients was 55.0 months (34.0–79.0 months). A majority of the patients were seen in the Midwest region of the United States (49.4%), followed by the Southern region (25.2%), which aligned with the territory covered by the Optum EHR.

### Model performance

The AuROC and DCA curves for the models are shown in Fig 2. The AuROC was 0.64 (95% CI: 0.64 to 0.64) for the LR-Demo model, 0.65 (95% CI: 0.65 to 0.66) for the LR-DemoLab model, and 0.80 (95% CI: 0.80 to 0.81) for the RF-DemoLab model. The addition of laboratory data slightly increased the overall AuROC, suggesting minimal discriminatory ability over demographic data alone. The RF-DemoLab model had a higher AuROC than LR-Demo or LR-DemoLab model, suggesting better discriminatory ability than the logistic models. DCA analysis suggests the RF-DemoLab model appears to have better net benefit when compared with the logistic models and intervention for all and intervention for none scenarios across a wide range of thresholds (Fig 2B).

Table 2 contains bootstrapped model performance metrics. For the LR-Demo model, the sensitivity was 68%, indicating reasonable ability to predict flare events in the following 6 months of a visit, but specificity was no better than chance at 51%. The LR-DemoLab model had similar performance: sensitivity of 69% and specificity of 54%. The RF-DemoLab model had the highest sensitivity of 74% and specificity of 72% and the lowest Brier score, indicating that this model had better diagnostic accuracy compared to the logistic regressions. The positive predictive value of 27% was highest for the RF-DemoLab (Table 2).

### Interpretability of the RF models

The 10 most important predictors based on TreeSHAP are presented in Fig 3. These included number of previous flares, age, mean potassium, white blood cell (WBC) count at visit and

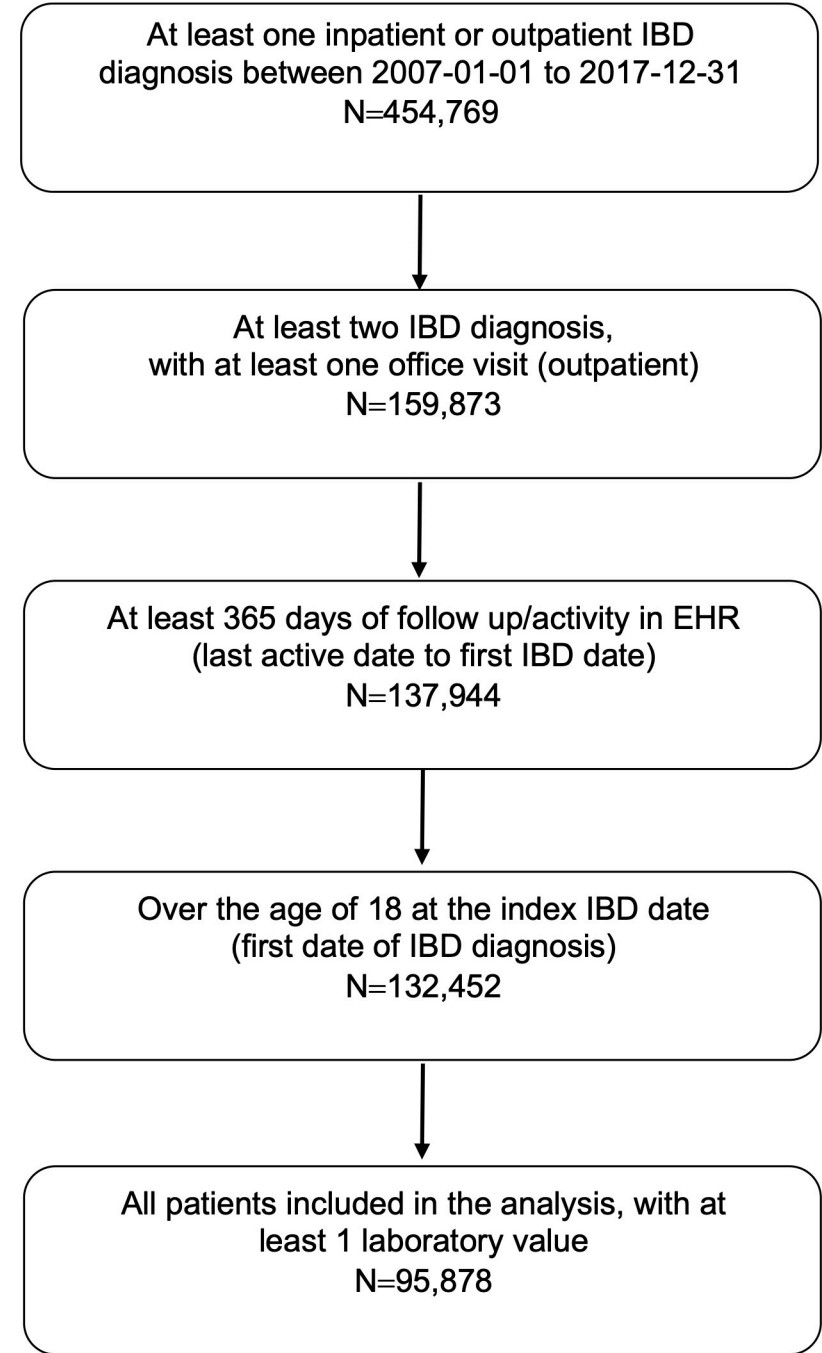

**Fig 1. Flow chart of identification of patients with inflammatory bowel disease (IBD) from the Optum electronic health records (EHR) database.**

past visit mean and maximum, calcium past visit mean, mean corpuscular volume (MCV) at current visit, blood urea nitrogen (BUN) past visit mean, and platelet past visit maximum.

TreeSHAP summary plot shows how each of these top 10 variables are influencing the log odds. For example, the variable for the number of previous flares in a patient shows that the log odds increase (positive SHAP value) for higher numbers of previous flares. For the variable of age, the log odds increase (positive SHAP value) for lower age values.

**Table 1. Patient characteristics.**

| Characteristic | All Patients | No Flare Event [a] | Had Flare Event |
|---|---|---|---|
| Number of patients (%) | 95,878 (100) | 73,633 (76.8) | 22,245 (23.2) |
| Age in years, mean (SD) | 48.2 (16.8) | 48.6 (16.6) | 47.1 (17.3) |
| Female, n (%) | 54,794 (57.1) | 41,645 (56.6) | 13,149 (59.1) |
| Deceased, n (%) | 3,304 (3.4) | 2,717 (3.7) | 587 (2.6) |
| Immunosuppressive medication use [b], n (%) | 20,375 (21.3) | 13,577 (18.4) | 6,798 (30.6) |
| Race, n (%) | | | |
| Black | 5,958 (6.2) | 4,291 (5.8) | 1,667 (7.5) |
| Asian | 998 (1.0) | 824 (1.1) | 174 (0.8) |
| White | 84,055 (87.7) | 64,532 (87.6) | 19,523 (87.8) |
| Other/Unknown | 3,876 (4.0) | 2,896 (3.9) | 980 (4.4) |
| Region, n (%) | | | |
| Midwest | 47,371 (49.4) | 36,274 (49.3) | 11,097 (49.9) |
| Northeast | 12,518 (13.1) | 9,845 (13.4) | 2,673 (12.0) |
| South | 24,198 (25.2) | 18,528 (25.2) | 5,670 (25.5) |
| West | 7,915 (8.3) | 6,090 (8.3) | 1,825 (8.2) |
| Disease, n (%) | | | |
| Crohn's disease | 42,977 (44.8) | 31,990 (43.4) | 10,987 (49.4) |
| Ulcerative colitis | 40,167 (41.9) | 33,483 (45.5) | 6,684 (30.0) |
| Indeterminate disease | 12,734 (13.3) | 8,160 (11.1) | 4,574 (20.6) |
| Number of non-flare visits [c] | | | |
| Mean (SD) | 7.2 (10.5) | 6.0 (8.3) | 11.4 (15.1) |
| Median (IQR) | 4.0 (2.0, 8.0) | 3.0 (2.0, 7.0) | 7.0 (3.0, 15.0) |
| Follow-up in months, mean (SD) | 58.5 (30.5) | 57.0 (30.2) | 63.4 (31.1) |
| Follow-up in months, median (IQR) | 55.0 (34.0, 79.0) | 53.0 (32.0, 77.0) | 61.0 (38.0, 85.0) |

IQR = interquartile range.

[a] Flare event was classified as an IBD-related hospitalization or corticosteroid prescription within 6 months of any inpatient or outpatient visit between 2007 and 2018.

[b] Immunomodulators and/or anti-tumor necrosis factor agents.

[c] Visits without a hospitalization or corticosteroid prescription in the following 6 months.

For comparison, the previous flare sum, age, potassium mean of past laboratory values, WBC count mean and maximum of past laboratory values were also identified as important variables in the variable importance plots (S1 Fig). Some differences are expected as variable importance by Gini impurity has some bias when variables with both multiple categories and continuous values are present [25].

TreeSHAP conditional dependency plots were created for the top 4 variables identified in the TreeSHAP summary plots were to describe the conditional non-linear relationship between the likelihood of a flare in the next 6 months and presented in S2 Fig. For comparison, odds ratios from the 2 logistic regression models are presented in S3 Fig.

## Sensitivity analyses

Sensitivity analysis for UC, CD, and IC subgroups are presented in the AuROC and RF variable importance S4 and S5 Figs, and were similar across patients with UC, CD and IC. This suggests that the trained RF-DemoLab model had similar predictive performance and variables regardless of diagnosis.

Performance metrics for the sensitivity analysis using the RF model using MICE imputation for laboratory values rather than simple median imputation are presented in S2 Table.

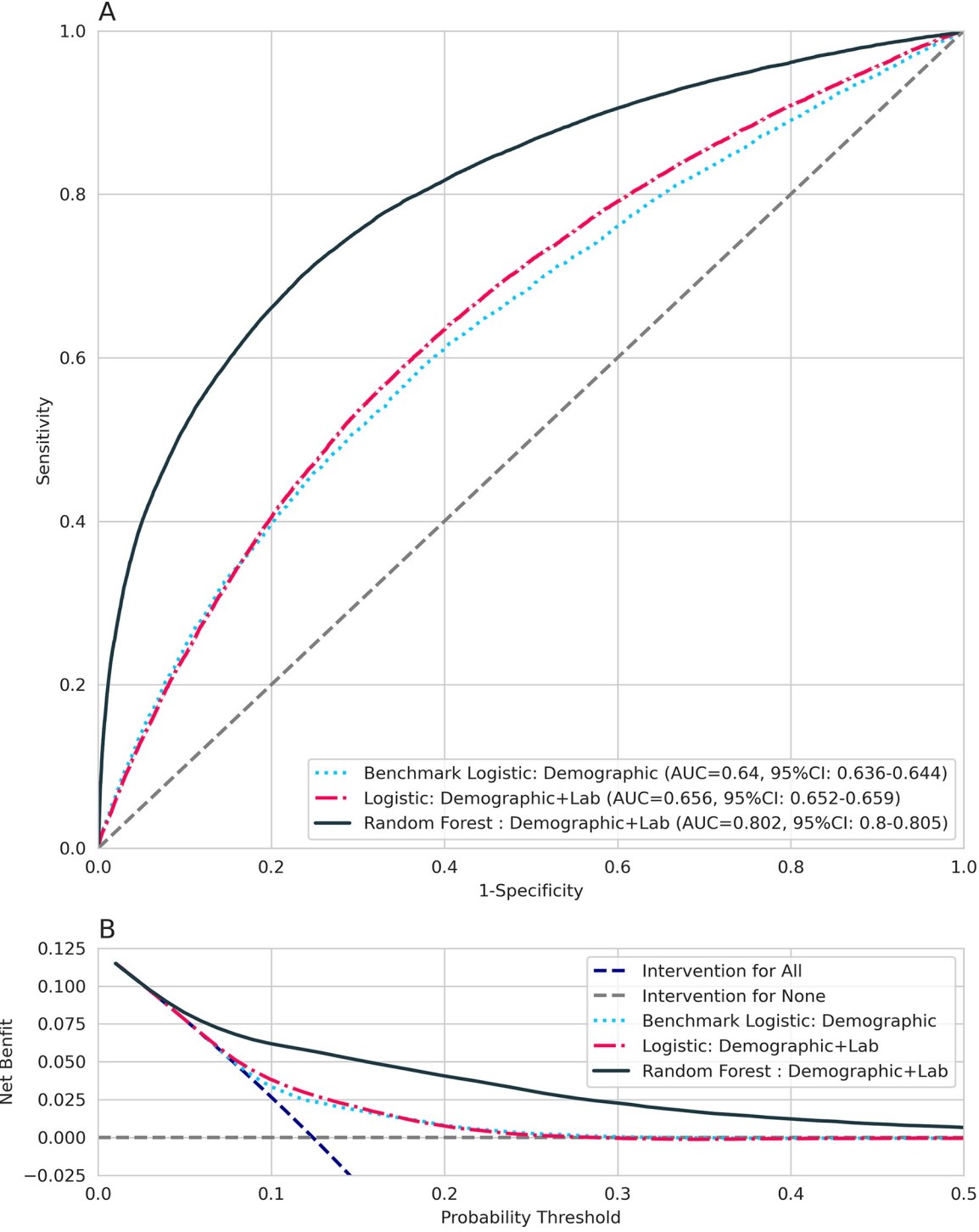

**Fig 2.** A) AuROC and B) DCA for prediction of flare in the next 6 months for the logistic regression model using demographic data, logistic regression using demographic and laboratory data, and random forest model using demographic and laboratory data. AUC = area under the curve; auROC = area under the receiver operating curve; DCA = decision curve analysis; RF = random forest model; ROC = receiver operating curve.

**Table 2. Bootstrapped estimates of median and 95% confidence interval (CI) model performance metrics.**

| Diagnostic Metric | LR-Demo | LR-DemoLab | RF-DemoLab |
|---|---|---|---|
| Sensitivity, median (95% CI) | 0.68 (0.67–0.69) | 0.69 (0.68–0.70) | 0.74 (0.73–0.74) |
| Specificity, median (95% CI) | 0.51 (0.51–0.51) | 0.54 (0.53–0.54) | 0.72 (0.72–0.72) |
| Positive predictive value, median (95% CI) | 0.16 (0.16–0.17) | 0.17 (0.17–0.18) | 0.27 (0.27–0.28) |
| Negative predictive value, median (95% CI) | 0.92 (0.92–0.92) | 0.92 (0.92–0.93) | 0.95 (0.95–0.95) |
| Accuracy, median (95% CI) | 0.53 (0.53–0.53) | 0.56 (0.55–0.56) | 0.72 (0.72–0.73) |
| Area under the curve, median (95% CI) | 0.64 (0.64–0.64) | 0.66 (0.65–0.66) | 0.80 (0.80–0.81) |
| Brier score | 0.26 | 0.26 | 0.19 |

LR-Demo = a logistic regression model using demographic data; LR-DemoLab = a logistic regression model using demographic and laboratory data; RF-DemoLab = a RF model using demographic and laboratory data.

Sensitivity in the RF-DemoLab model using MICE imputation was lower (0.60) compared to the RF-DemoLab model with simple median imputation (0.74), resulting in lower accuracy overall. This suggests that linear iterative imputers such as MICE may not be ideal for tree-based modeling tasks.

## Discussion

The model-building process was replicated to predict flare in patients with IBD using the Optum EHR with similar predictive accuracy as Waljee et al. [3] and following TRIPOD guidelines [24]. The RF model with both clinical and laboratory variables saw improvements in all predictive accuracy metrics and DCA net benefit when compared to the logistic models. This higher performance is likely due to the ability of the RF algorithm to implicitly handle interactions between variables as well as non-linearities, which must be explicitly modeled in logistic regression.

Results from the TreeSHAP algorithm offer insight and model interpretability to RF models. The number of previous flare events a patient experienced was found to be the most important variable at predicting a flare in the next 6 months. This relationship showed a threshold effect where the influence of previous flares on the likelihood of future flares increased linearly as the number of previous flares increased up to 3 previous flares and remained elevated at > 3 previous flares (S2A Fig).

While the process outlined in Waljee et al. [3] was generally replicable and reproducible, performance metrics in this study were noticeably lower. In Waljee et al., the AuROC for the RF model with clinical variables (including previous flare) and laboratory variables was 0.87 [3], where this comparable model had an AuROC of 0.80. One explanation for the lower performance metrics observed could be the difference in the patient populations. Where this study evaluated patient populations with IBD in the commercial Optum EHR, Waljee et al. [3] evaluated patients with IBD in the VHA. The VHA population was predominantly male and had an average age approximately 10 years older than the Optum EHR cohort. In addition, our study required 12 months of post-baseline follow-up data, which would limit model generalizability to only to patients with continuous care within a single healthcare system.

This study deviated from some preprocessing of laboratory values when compared to Waljee et al. [3] Specifically, laboratory variables were required to be measured on at least 70% of visits, and a median imputation of population values rather than a past median per patient was used. It is possible that these criteria also resulted in lower model performance. However, these criteria were made to simplify the preprocessing of laboratory data, which made interpretations of model SHAP values and odds ratios easier to understand. Furthermore, highly

correlated lab values such as WBC past mean and WBC past max values, could result in some dilution of variable importance in both SHAP (Fig 3) and VIF (S1 Fig). Future work could use feature selection approaches to reduce the number of correlated variables.

The use of the Optum EHR may have introduced misclassification. Oral corticosteroids received during the course of a hospitalization were not available and inpatient care received outside the Optum network could not be tracked. Clinical information, such as disease extent and location, were also not captured in the Optum EHR. As a result, the analysis of the Optum EHR reported here is anticipated to have more missing data than that reported by Waljee et al. [3] These limitations may explain the lower proportion of events in this analysis and subsequent lower model performance metrics.

Another difference between this study and Waljee et al. [3] was that patients with concurrent Clostridium difficile were not excluded. While this may have led to misclassification, a positive C. difficile diagnosis was observed in < 2% of the study population. In addition, objective definitions of flare, such as using fecal calprotectin [26], were not specifically examined. Specifically, fecal calprotectin was measured on < 1% of the study population. Measurements from endoscopy and disease specific activity/severity scores were also not available in the EHR. Future research could work on disease-specific models integrating fecal calprotectin and other UC- and CD-specific disease activity/severity scores with clinical and laboratory variables readily available in EHR.

Reproduction of a study is essential for replication of scientific findings. However, this is a challenge in biomedical sciences, as often not enough key elements to reproduce a study are made available to external researchers, including detailed methods, data, and code [27]. Data sharing is not always feasible in biomedical research due to data use agreements and privacy regulations, as was the case in this study which prohibited external validation of the trained models on the VHA cohort. However, other essential elements of reproducible research can be provided, including use of open source software and sharing code, to increase likelihood of replicating a study [28]. These elements, in addition to a detailed methods section, helped in replicate the findings of Waljee et al. [3]

Machine learning models could help enable personalized care in IBD patients and could be incorporated into routine practice as clinical decision support tools for monitoring and

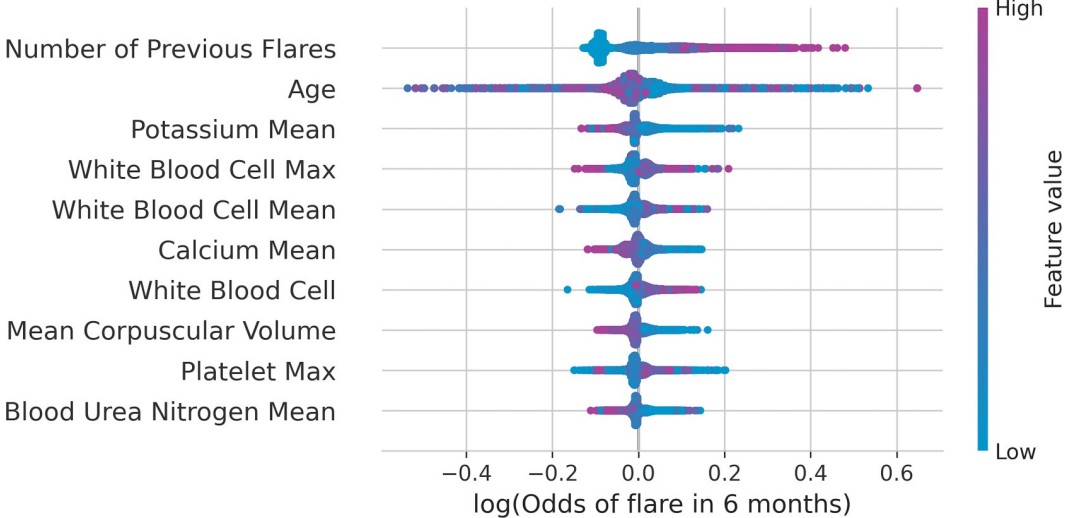

**Fig 3. TreeSHAP summary plot of the top 10 most important variables for predicting a flare in the next 6 months for patients with IBD.** IBD = inflammatory bowel disease; Max = maximum.

identifying patients at high-risk of flare. As algorithms such as RF are utilized more in clinical decision-making, it will be important for patients or clinicians to be able to make sense of algorithms beyond high level accuracy metrics. Use of SHAP values can help accomplish this task. However, model interpretability is only one aspect. Ensuring machine learning models are validated in different EHRs and generalizable will be critical to promote use in routine practice.

## Supporting information

**S1 Fig. Random forest top 10 features estimated using variable importance.**
IBD = inflammatory bowel disease; Max = maximum.
(TIFF)

**S2 Fig. Treeshap dependency plot of the top 4 important variables.** Dependency Plot For A) Number of Previous Flares, B) Age At Visit, C) Potassium Mmol/L Past Visits Mean, and D) WBC Count×10$^3$/μL Past Visits Mean.
(TIFF)

**S3 Fig. Odds ratios from logistic regression prediction models for the demographics only model, and the demographics and laboratory variables.** Max = maximum; Labs = laboratory variables. Note: Python scikit-learn used for logistic models does not estimate variance parameters for the coefficients and are therefore not reported.
(TIFF)

**S4 Fig. AuROC for random forest (RF) models (clinical and laboratory variables) subset by ulcerative colitis, Crohn's disease, and indeterminate colitis.** AUC = area under the curve; auROC = area under the receiver operating curve; RF = regression model; ROC = receiver operating curve.
(TIFF)

**S5 Fig. Random forest top 10 features estimated using variable importance subset by ulcerative colitis, Crohn's disease, and indeterminate colitis.** Max = maximum.
(TIFF)

**S1 Table. Patient variables.**
(DOCX)

**S2 Table. Bootstrapped estimates of median and 95% Confidence Interval (CI) model performance for random forest (RF) model with clinical and multiple imputation by chain equation laboratory features on laboratory variables measured at >70% of visits.**
(DOCX)

**S3 Table. TRIPOD checklist for prediction model development and validation.**
(DOCX)

**S1 List. Generic names of corticosteroids.**
(DOCX)

**S2 List. ICD-9-CM codes for a variety of common inflammatory comorbid conditions.**
(DOCX)

## Author Contributions

**Conceptualization:** Ryan W. Gan, Diana Sun, Amanda R. Tatro, Shirley Cohen-Mekelburg, Wyndy L. Wiitala, Ji Zhu, Akbar K. Waljee.

**Data curation:** Ryan W. Gan.

**Formal analysis:** Ryan W. Gan.

**Funding acquisition:** Diana Sun.

**Methodology:** Ryan W. Gan, Wyndy L. Wiitala, Ji Zhu.

**Project administration:** Akbar K. Waljee.

**Supervision:** Amanda R. Tatro, Akbar K. Waljee.

**Visualization:** Ryan W. Gan.

**Writing – original draft:** Ryan W. Gan, Diana Sun, Amanda R. Tatro, Shirley Cohen-Mekelburg, Wyndy L. Wiitala, Ji Zhu, Akbar K. Waljee.

**Writing – review & editing:** Ryan W. Gan, Diana Sun, Amanda R. Tatro, Shirley Cohen-Mekelburg, Wyndy L. Wiitala, Ji Zhu, Akbar K. Waljee.

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
