## [Decision Letter · Decision Letter 0]

26 Jul 2021

PONE-D-21-20616

REPLICATING PREDICTION ALGORITHMS FOR HOSPITALIZATION AND CORTICOSTEROID USE IN PATIENTS WITH INFLAMMATORY BOWEL DISEASE

PLOS ONE

Dear Dr. Waljee,

Thank you for submitting your manuscript to PLOS ONE. After careful consideration, we feel that it has merit but does not fully meet PLOS ONE’s publication criteria as it currently stands. Two expert reviewers in this field identified some area need further revision. Therefore, we invite you to submit a revised version of the manuscript that addresses the points raised during the review process.

We look forward to receiving your revised manuscript.

Kind regards,

Hsu-Heng Yen

Academic Editor

PLOS ONE

Journal Requirements:

2. Please note that according to our submission guidelines (http://journals.plos.org/plosone/s/submission-guidelines), outmoded terms and potentially stigmatizing labels should be changed to more current, acceptable terminology. For example: “Caucasian” should be changed to “white” or “of [Western] European descent” (as appropriate).

[I have read the journal's policy and the authors of this manuscript have the following competing interests:

Ryan Gan and Diana Sun are full time employees of Genentech, Inc., a member of the Roche group, and own shares of Roche stock.

Amanda Tatro is a full time employee of F. Hoffmann La Roche AG and own shares of Roche stock.].    

We note that one or more of the authors are employed by a commercial company: Roche stock. 

Reviewers' comments:

Reviewer's Responses to Questions

**Comments to the Author**

1. Is the manuscript technically sound, and do the data support the conclusions?

Reviewer #1: Yes

Reviewer #2: Partly

2. Has the statistical analysis been performed appropriately and rigorously? 

Reviewer #1: Yes

Reviewer #2: Yes

3. Have the authors made all data underlying the findings in their manuscript fully available?

Reviewer #1: Yes

Reviewer #2: Yes

4. Is the manuscript presented in an intelligible fashion and written in standard English?

Reviewer #1: Yes

Reviewer #2: Yes

5. Review Comments to the Author

Reviewer #1: Thank you for the opportunity to review this manuscript. Statistical analysis provided accurate data to explain predictors of IBD flare. It will supply clinicians useful information in decision-making.

Reviewer #2: Previously, this group developed a machine-learning model that more accurately predicts IBD flares. Corticosteroid use and previous hospitalization were key variables of the model (Waljee et al 2018). The present study aim to replicate and validate the model previously published using a distinct and independent IBD cohort, which is clinically relevant. This study could be improved by attention to some concerns that are described below.

General comments

1) Predictor variables: fecal calprotectin was not available, what about HBI (CD), Mayo score (UC), Ulcerative Colitis Disease Activity Index (UCDAI), endoscopy scores? Baseline and overtime disease activity/severity is a more accurate tool to determine real flares.

2) Why diagnosis variable (CD, UC, IC) were not included in the prediction models? Since CD and UC are two distinct diseases, with distinct pathologies and characteristics and given the large sample size here used, independent analyses based on disease type would better support the generalizability of the model prediction for IBD flares.

3) Abstract is lacking sensitivity and specificity results.

4) Cross-validation analysis was used to provide evidence for generalizability. It does provide internal validation, but a more robust approach would be to train the data on the previous published VHA cohort and test on the present cohort (exteran validation).

Specific comments

1) Abstract conclusion and key words have a different format.

2) Introduction: ”they found a random forest (RF) model improved the ability to predict IBD flares and outperformed logistic regression models”. Grammatically confuse.

3) Introduction: “(1) to replicate the findings of models developed by Waljee et al.[3] using demographic and laboratory data in independent dataset of a commercial EHR to predict hospitalization and corticosteroid use as a surrogate for IBD flares in a community-based cohort;”. Statement is too long and grammatically confuse.

4) Study population: “and were observed from the index date until the last observed observation”?

5) Figures: bad quality/resolution.

6. PLOS authors have the option to publish the peer review history of their article (what does this mean?). If published, this will include your full peer review and any attached files.

Reviewer #1: No

Reviewer #2: **Yes: **Svetlana Ferreira Lima

---

## [Author Response · Author response to Decision Letter 0]

25 Aug 2021

Response to edits requested on submission PONE-D-21-20616R1: 

We've checked your submission and before we can proceed, we need you to address the following issues:

1. Thank you for stating the following in the Competing Interests section: 

[I have read the journal's policy and the authors of this manuscript have the following competing interests:

Ryan Gan and Diana Sun are full time employees of Genentech, Inc., a member of the Roche group, and own shares of Roche stock.

Amanda Tatro is a full time employee of F. Hoffmann La Roche AG and own shares of Roche stock.]. 

We note that one or more of the authors are employed by a commercial company: Roche stock. 

-Amended Funding Statement provided in updated Cover Letter. The funding organization only provided financial support in the form of authors’ salaries and/or research materials and is reflected as such in the Author Contributions. 

-This statement has been included in our amended Funding Statement within the updated Cover Letter. 

-We have updated our Competing Interests statement in our Cover Letter adding the following statement “This does not alter our adherence to PLOS ONE policies on data sharing and materials.”

2. Please remove your Supporting Information figures within your manuscript file, leaving only the individual TIFF/EPS image files. These will be automatically included in the reviewer’s PDF

-We have removed Supporting Information figures within the manuscript file, only leaving the figure captions.

3. We note your current Data Availability statement is:

"The Optum EHR [5] data used in this study were licensed from Optum and are not publicly available due to data licensing and use agreements; interested researchers can contact Optum to license the data. All interested researchers can access the data in the same manner as the authors. The authors had no special access privileges. Optum EHR contact website: https://www.optum.com/business/solutions/government/federal/data-analytics-federal/clinical-data.html.

R code (GitHub: https://github.com/CCMRcodes/IBD_Flare) and paper previously published by Waljee et al.[3] was reviewed and adapted to Python code.

Manuscript code and summary data used to produce tables and plots can be found in the public GitHub repository: https://github.com/phcanalytics/ibd_flare_model. Note, SHAP values require raw data to be calculated and are not included in the repository due to data use agreements.

Transparent Reporting of a Multivariable Prediction Model for Individual Prognosis or Diagnosis (TRIPOD) guidelines were implemented;[24] checklist can be found in S3 Table."

PLOS defines the "minimal data set" as consisting of the data used to reach the conclusions drawn in the manuscript with related metadata and methods, and any additional data required to replicate the reported study findings in their entirety. This includes:

Before we proceed, please address the following prompts:

a.) Please confirm whether your current Data Availability statement includes all of the relevant information and links needed for accessing the minimal data set necessary for replicating the study's results?

-We confirm that our Data Availability statements includes all relevant information and links need for accessing the minimal data set necessary for replicating the study’s results. In an effort to improve data accessibility, we have put a markdown table on meta data used to make the plots and tables in the results folder on the public git repository and modified the data availability paragraph slightly to separate out the link for the summary data availability. We have updated our Data Availability statement in the manuscript slightly to read “ The Optum EHR [5] data used in this study were licensed from Optum and are not publicly available due to data licensing and use agreements; interested researchers can contact Optum to license the data. All interested researchers can access the data in the same manner as the authors. The authors had no special access privileges. Optum EHR contact website: https://www.optum.com/business/solutions/government/federal/data-analytics-federal/clinical-data.html.

R code (GitHub: https://github.com/CCMRcodes/IBD_Flare) and paper previously published by Waljee et al.[3] was reviewed and adapted to Python code. 

Manuscript code used to produce tables and plots can be found in the public GitHub repository: https://github.com/phcanalytics/ibd_flare_model. Summary data used to create plots and tables can be found in the public GitHub repository: https://github.com/phcanalytics/ibd_flare_model/tree/master/results. Note, SHAP values require raw data to be calculated and are not included in the repository due to data use agreements. 

Transparent Reporting of a Multivariable Prediction Model for Individual Prognosis or Diagnosis (TRIPOD) guidelines were implemented;[24] checklist can be found in S3 Table.”

---

## [Editor Report · Decision Letter 1]

6 Sep 2021

REPLICATING PREDICTION ALGORITHMS FOR HOSPITALIZATION AND CORTICOSTEROID USE IN PATIENTS WITH INFLAMMATORY BOWEL DISEASE

PONE-D-21-20616R1

Dear Dr. Waljee,

We’re pleased to inform you that your manuscript has been judged scientifically suitable for publication and will be formally accepted for publication once it meets all outstanding technical requirements.

Kind regards,

Hsu-Heng Yen

Academic Editor

PLOS ONE
---

## [Editor Report · Acceptance letter]

10 Sep 2021

PONE-D-21-20616R1 

Replicating prediction algorithms for hospitalization and corticosteroid use in patients with inflammatory bowel disease 

Dear Dr. Waljee:

I'm pleased to inform you that your manuscript has been deemed suitable for publication in PLOS ONE. Congratulations! Your manuscript is now with our production department. 

Kind regards, 

on behalf of

Dr. Hsu-Heng Yen 

Academic Editor

PLOS ONE